# Conversion of Pulse Protein Foam-Templated Oleogels into Oleofoams for Improved Baking Application

**DOI:** 10.3390/foods11182887

**Published:** 2022-09-17

**Authors:** Athira Mohanan, Kim Harrison, David M. L. Cooper, Michael T. Nickerson, Supratim Ghosh

**Affiliations:** 1Department of Food and Bioproduct Sciences, University of Saskatchewan, Saskatoon, SK S7N 5A8, Canada; 2Department of Anatomy Physiology and Pharmacology, University of Saskatchewan, Saskatoon, SK S7N 5E5, Canada

**Keywords:** foam-templated oleogelation, pulse proteins, oleofoam, cake texture profile, monoacylglycerol, rheology, X-ray microtomography

## Abstract

The food industry has long been searching for an efficient replacement for saturated-fatty-acid-rich fats for baking applications. Although oleogels have been considered a potential alternative for saturated and trans fats, their success in food application has been poor. The present study explored the use of oleofoams obtained by whipping the pulse protein foam-templated oleogels for cake baking. Oleogels were prepared at room temperature by adding canola oil containing high-melting monoglyceride (MAG) or candelilla wax (CW) to the freeze-dried pea or faba bean protein-stabilized foams. Oleogels were then whipped to create the oleofoams; however, only the oleogels containing MAG could form oleofoams. CW-oleogel could not form any oleofoam. The most stable oleofoams with the highest overrun, stability, and storage modulus were obtained from 3% MAG+pulse protein foam-templated oleogels. The MAG plus protein foam-templated oleogels showed smaller and more packed air bubbles than MAG-only oleofoam, which was ascribed to the protein’s ability to stabilize air bubbles and provide a network in the continuous oil phase to restrict air bubble movement. A novel batter preparation method for oleofoam was developed to increase air bubble incorporation. The X-ray microtomography images of the cakes showed a non-homogeneous distribution of larger air bubbles in the oleofoam cake compared to the shortening cake although their total porosity was not much different. The oleofoam cakes made with the new method yielded similar hardness and chewiness compared to the shortening cakes. By improving rheology and increasing air incorporation in the batter, high-quality cakes can be obtained with MAG-containing oleofoams made from pulse protein foam-templated oleogels.

## 1. Introduction

Oleogels have been considered as a potential alternative for saturated and trans fats in various foods, including baked products, processed meats, meat, and dairy alternatives [1,2]. The latest trend in oleogel preparation is using food hydrocolloids such as proteins and polysaccharides or their combination as oleogelators [3,4,5,6] instead of conventional oleogelators such as plant waxes, monoglycerides, fatty acids, and ethyl cellulose [7,8,9]. Water-soluble proteins and polysaccharides, however, cannot make networks capable of holding liquid oil if directly added to the oil. Therefore, indirect approaches that facilitate the biopolymer network formation capable of holding liquid oil are required. In the indirect approach, hydrogels [4,5,10], foams [11], or emulsions [12,13] were stabilized using a biopolymer or combinations of biopolymers, followed by the removal of water to obtain the polymer network capable of holding the liquid oil.

Proteins possess higher nutritional benefits over other oleogelators. Several studies have already shown that the proteins can be successfully utilized to convert liquid oils into oleogels via indirect approaches. de Vries et al. [4] effectively converted whey protein stabilized hydrogels into oleogels by stepwise solvent exchange. A combination of gelatin and xanthan gum (XG) was used to prepare oleogels via the foam-templated approach [14], where the biopolymers-stabilized aqueous foams were freeze-dried to obtain a network capable of trapping liquid oil and facilitate the formation of oleogels. Mohanan et al. [6] were the first to prepare oleogels using pulse proteins from pea and faba bean in combination with XG via the foam-templated approach. However, one of the limitations of these pulse protein-stabilized oleogels was their poor oil-binding capacity. In a subsequent study, Mohanan et al. [15] showed that the addition of a small concentration (<3%) of high-melting monoacylglycerols (MAG) or candelilla wax (CW) in the oil phase can significantly improve the oil-binding capacity of pulse protein-based oleogels.

Despite their success in simulating structural properties of conventional fats (rich in saturated fatty acids), hydrocolloid-stabilized oleogels have not shown great success when incorporated into various food applications [8,15,16]. Although cakes were successfully baked using these oleogels, the quality of the batters and the cakes prepared using shortening was superior to the oleogel-based batters and cakes. Cake batters produced using hydroxypropyl methylcellulose (HPMC) and pulse protein-based oleogels displayed higher specific gravity and lower viscosity and viscoelasticity than the corresponding shortening batter [6,16]. These were found to be due to the lower air incorporation in oleogel batters due to the inability of the oleogels to stabilize air bubbles, similar to shortening, where fat crystals were mostly responsible for the air bubble stabilization. Poor air incorporation in the oleogel batters compared to the shortening batter resulted in cakes with higher hardness and chewiness than the corresponding shortening cakes. Mohanan et al. [6] have shown that the cakes prepared using pulse protein-stabilized oleogels were not so different from canola-oil-based cakes. Their study also revealed that the inability of oleogels to mimic shortening was due to the destabilization of oleogels during the batter preparation and release of the liquid oil. Although adding small concentrations of high-melting MAG and CW increased the viscoelasticity and oil-binding capacity of oleogels, during cake baking, only the oleogel consisting of MAG was able to improve the quality of the cakes [15]. This was due to the ability of MAG to incorporate more air into the cake batter due to its surface activity and crystal formation. Similarly, for shortening, the presence of surface-active mono- and diacylglycerols (MAG and DAG) and solid fat crystals from high-melting triacylglycerols helped to stabilize more air into the cake batter. This shows that the incorporation and stabilization of air bubbles in the cake batter is essential to producing high-quality cakes or other aerated bakery products.

The present study examines how to improve the air incorporation in the batter and the quality of the cakes baked with oleogels by converting the oleogels into oleofoams before cake baking. Conventional oleofoams are air bubbles dispersed in a continuous oil medium, where the air bubbles are stabilized by the Pickering action of surface-active fat crystals such as MAG [17]. One of the first reports of oleofoam developed from fatty alcohol-based oleogels was made by Fameau et al. [18], who showed that fatty alcohol crystals were responsible for Pickering stabilization of air bubbles in a continuous matrix of crystal network in liquid sunflower oil. Later Truong et al. [19] found that the oleogels prepared using a blend of MAG and native phytosterols (10 wt%) in canola oil can be converted into oleofoams by whipping. The oleofoams were stable for months, and their stability and viscoelastic properties could be controlled by modulating the oleogel preparation conditions. The author showed that only the oleogels prepared at a faster cooling rate leading to smaller fat crystals could form oleofoams by the adsorption of fat crystals on the air bubbles [19]. The high long-term stability of the oleofoams was attributed to both the interfacial stability of the crystals and the bulk rheology of the crystal network leading to the prevention of both the Ostwald ripening and bubble destabilization [20]. A few patents on the application of oleofoams in confectionary products have recently been published [21,22]; however, no detailed report on the mechanism of action and the use of oleofoams in bakery applications could be found. In the present study, oleogels prepared using faba bean and pea protein concentrates along with MAG and CW were used to develop stable oleofoam to test their application in cake baking. The overall goal was to investigate how much of the conventional high-melting shortening functionality in cake baking can be matched by the pulse protein plus MAG- or CW-stabilized oleofoams.

## 2. Materials and Methods

### 2.1. Materials

Faba bean protein concentrate (FPC) and pea protein concentrate (PPC) were kindly donated by AGT Food and Ingredients (Saskatoon, SK, Canada). FPC contained 57.6% protein, 6.2% moisture, 1.6% lipid, 5.4% ash, and the rest 29.2% carbohydrates, while PPC contained 51.4% protein, 6.8% moisture, 2.1% lipid, 5.4% ash, and the rest 34.3% carbohydrates [15]. Gluten-free xanthan gum (XG) was purchased from Bulk Barn store, a local supplier of Duinkerken Foods Inc. (Slemon Park, PE, Canada). Vegetable shortening (Crisco brand, composed of soybean oil, hydrogenated palm oil, modified palm oil, mono and diglycerides, TBHQ, and citric acid), and canola oil (Great Value brand) were purchased from Walmart Supercentre (Saskatoon, SK, Canada). Refined candelilla wax was donated by Multiceras (Monterrey, NL, Mexico). Powdered monoacylglycerol (MAG, product code DMG0093) was donated by Palsgaard (Palsgaard Industry de Mexico, San Luis Potosi, Mexico), which is a mixture of monoacylglycerols contained about 37% glycerol mono-stearate, 54% glycerol monopalmitate, and 7.5% free fatty acids with a melting point 70 °C. Deionized water (Millipore) was used for all the solution preparations. Sodium azide and all other chemicals were purchased from Sigma Aldrich Canada (Oakville, ON, Canada).

### 2.2. Oleogel Preparation

Pulse-protein-stabilized oleogels were prepared using a foam-templated approach. Stable foams were first prepared using a mixture of 5 wt% protein (PPC or FPC) and 0.25 wt% of XG at pH 7, according to Mohanan et al. [23]. Briefly, 400 mL proteins with XG solutions were whipped using a KitchenAid Ultra Power Mixer (Whirlpool Canada LP, Mississauga, ON, Canada) using a 4.5 qt (4.3 L) stationary bowl and stainless-steel rotating beaters at a speed setting of 8 (380 rpm) for 20 min. These foams were immediately transferred to a 20 cm × 20 cm aluminum tray and frozen at −30 °C for 24 h. The frozen foams were then dried for 72 h using a freeze dryer (Labconco FreeZone 18 Liter Console Freeze Dryers, Labconco Corp, Kansas City, MO, USA).

Oleogels were prepared by adding a hot mixture (80 °C) of canola oil containing 0–3% CW or MAG into ~7 g of freeze-dried foams in a 500 mL glass beaker. The oil mixture was prepared by dissolving the required concentration of CW or MAG in canola oil at 80 °C. The addition of the oil mixture to the dry foam stopped when the weight of the added oil was 30 times the foam weight. No shearing was applied, as we previously reported that shearing at this stage would damage the freeze-dried foam structure [6,15]. The beakers were then quickly transferred to a refrigerator (4 °C), a freezer (−20 °C), or left at room temperature (22 °C) to allow the formation of oleogel. The samples were left in the desired storage condition for 24 h before further usage. As a control, a hot mixture of canola oil containing different concentrations of CW or MAG (without protein foams) was also added to beakers and left under the same storage conditions. The freeze-dried protein foams with just canola oil (without any CW or MAG) were also kept under the same storage conditions as another control.

### 2.3. Oleofoam Preparation and Characterization

Oleofoams were prepared by whipping 200 g of oleogels using a KitchenAid Ultra Power Mixer (Whirlpool Canada LP, Mississauga, ON, Canada) using a 4.5 qt (4.3 L) stationary bowl and stainless-steel rotating beaters at a speed setting of 10 (415 rpm) for 30 min.

#### 2.3.1. Overrun and Oleofoam Stability

The foamability of the oleofoam was obtained by measuring the overrun immediately after whipping, according to Mohanan et al. [23]. A part of the oleofoams was gently scooped out of the bowl using a spatula to fill two tared 50 mL measuring cylinders. The excess foam was scraped off the top of the cylinder using a spatula to obtain constant volume for each measurement. The weight of the oleofoam (*W*_50 *mL oleofoam*_) was recorded and used in the overrun calculation using Equation (1) [23].
(1)$$\%\;\mathrm{Overrun}\;=\frac{\;W_{50\;mL\;oleogel}-W_{50\;mL\;oleofoam}}{W_{50\;mL\;oleofoam}}*100$$
where *W*_50 *mL oleogel*_ is the weight of an equal volume of oleogel used for oleofoam preparation. Oleofoam stability was measured according to [24] with slight modifications. Freshly prepared oleofoams (50 mL) were placed into 50 mL graduated cylinders with the top covered with a parafilm and left at room temperature to drain liquid oil. The change in oleofoam volume was recorded as a function of time. Foam stability (%FS) was calculated from the percentage of the volume of foam left after six hours (*V_f_*) to the initial foam volume (*V_i_* = 50 mL) using Equation (2).
(2)$$\%FS=\frac{V_{f}}{V_{i}}*100$$

#### 2.3.2. Microstructure of Oleofoams

The microstructure of oleofoams was obtained using a bright field and polarized light microscope (Nikon Eclipse E400 microscope, connected with a Nikon DS-Fil camera) using a 10× objective lens at room temperature (25 ± 2 °C). The microscopy of the oleofoams was performed after 24 h of storage in the refrigerator.

#### 2.3.3. Viscoelasticity of Oleofoams

The viscoelastic properties of the oleofoams were determined within 5 min after their preparation to understand their elastic strength using an AR-G2 rheometer (TA Instruments, Montreal, QC, Canada). A 40 mm acrylic parallel plate geometry was used for viscoelasticity analysis. Oleofoams were gently loaded on the Peltier plate of the rheometer with a spatula. Excess foam coming out of the geometry was gently wiped off with a spatula. An oscillatory strain sweep (from 0.01% to 100%) was applied at a constant frequency of 0.5 Hz at 25 °C to find the linear viscoelastic region (LVR). Then, a frequency sweep measurement was performed from 0.01 to 100 rad/s angular frequencies at a constant strain of 0.05% within the LVR for most samples. The storage (G′) and loss modulus (G″) of the samples were recorded with the TRIOS Software version 4.0.2.30774 (TA Instruments, Montreal, QC, Canada).

### 2.4. Cake Baking with Oleogels and Oleofoams

Cakes were baked using a slightly modified AACC International Method 10–90.01 [25] and a lab-developed new method. While using the oleofoams, the AACC method did not give the best results due to the collapse of the oleofoam structure. Therefore, the AACC method was modified to minimize disruption in oleofoam structure in the cake batter while optimizing the final cake texture.

In both methods, batters were prepared using 200 g all-purpose flour, 280 g crystalline sugar, 100 g fat, 24 g non-fat dried milk, 18 g dried egg white powder, 6 g NaCl, 12.5 g baking powder, and 250 g water. The ingredients were mixed using a KitchenAid Ultra Power Mixer (Whirlpool Canada LP, Mississauga, ON, Canada) using a 4.5 qt (4.3 L) stationary bowl and rotating stirrers. The fat phase of the batter consisted of either 100% vegetable shortening, 100% canola oil, 100% oleogels, or 100% oleofoams prepared from FPC-XG + 3% MAG or PPC-XG + 3% MAG oleogels. As controls, oleogels and oleofoams prepared with only 3% MAG in canola oil and only the freeze-dried foams of PPC-XG or FPC-XG in canola oil (without MAG) were also used.

In the AACC method, all the dry ingredients were mixed before adding them to the mixing bowl. Then, the fat phase was added with 150 mL of water and mixed for 1 min at speed 2 followed by 4 min at speed 4 (275 rpm). The rest of the water was added in steps with various mixing speeds according to the AACC method. In the lab-developed new batter-preparation method, different mixing procedures and stepwise addition of water and oleofoam were explored to keep the oleofoam structure stable in the cake batters so that the specific gravity of the oleofoam batters would be as close as the shortening batter prepared using the AACC method. It was found that the only way to keep the oleofoam structure stable in the batter and prevent the formation of long gluten networks was to add the oleofoams stepwise. Based on several preliminary experiments, the final method was to mix all the dry ingredients before adding them into the mixing bowl, followed by the addition of half of the oleofoam (50 g) and 150 mL of water and mixed at speed 2 for 30 s and speed 4 for 1 min in the KitchenAid mixer. The rest of the water (100 mL) was then mixed at speed 2 for 1 min, followed by adding the remaining half of the oleofoam (50 g) and mixing at speed 2 for 1 min.

In both methods, approximately 200 g of dough were transferred to a baking tray (15.0 × 7.5 × 5.5 cm) and baked in an electric oven at 195 °C until done (AACC, 1999) by inserting a toothpick and making sure no crust was stuck to it. The time of baking varied between 20–25 min for all cakes. After baking, the cakes were cooled to room temperature for 30 min before removing them from the tray. Freshly baked cakes were covered with aluminum foil and plastic wrap to prevent moisture loss or gain until further analysis.

### 2.5. Characterization of Cake Batters and Cakes

#### 2.5.1. Specific Gravity of Cake Batter

The specific gravity of cake batters was determined from the ratio of the weight of 100 mL batter to the weight of the same volume of water [6,15].

#### 2.5.2. Microstructure of Cake Batter

The microstructure of the batters was obtained using a polarized light microscope with a 10X objective lens and a confocal laser scanning microscope with 10×, 40×, and 60× objective lenses at room temperature. Batters were analyzed within 30 min after preparation to minimize the effect of time. A small amount of batter was compressed gently on a glass slide with a coverslip, and the polarized light microstructure was assessed using a Nikon Eclipse E400 microscope (Nikon Canada Inc., Mississauga, ON, Canada) connected with a Nikon DS-Fil camera. Confocal laser scanning micrographs of cake batters were captured using a Nikon C2 microscope (Nikon Inc., Mississauga, ON, Canada) using a combination of 543 and 633 nm lasers and a 10× and 40× objective lens. Nile red (excitation by 543 nm laser, with emission collected in 573–613 range) and fast green (excitation by 633 nm laser, with emission collected using a 650 nm long-pass filter) were used to stain the oil phase and the proteins in the aqueous phase, respectively. For oleogel and oleofoam batters, 0.01 wt% Nile red was dissolved in canola oil before its addition to the dry foam, while for shortening batter, 0.01 g of Nile red was dissolved in 1 g canola oil and mixed with the shortening. To stain the proteins, a few drops of 0.1 wt% fast green solution in water was added to 150 mL of water used for batter preparation.

#### 2.5.3. Rheology of Cake Batter

The viscosity and viscoelasticity of cake batters were measured using the AR-G2 rheometer (TA Instruments, Montreal, QC, Canada) with a 40 mm acrylic parallel plate at 25 °C. Viscosity was determined as a function of shear rate from 0.01 to 1000 s^−1^ with a gap of 1000 µm between the two parallel plates. For viscoelasticity, an oscillatory strain sweep (from 0.01% to 100%) was applied at a constant frequency of 0.5 Hz at 25 °C to determine the linear viscoelastic region (LVR). Then, a frequency sweep measurement was performed from 0.01 to 100 rad/s at a constant strain of 0.05% within the LVR.

#### 2.5.4. X-ray Microtomography of Cake

Porosity within the cake was investigated by X-ray micro-computed tomography (CT) using a SkyScan 1172 (Aartselaar, Belgium) X-ray microtomograph (<5 μm X-ray source spot size; 8.83 camera pixel size) operated at 40 kV and 250 μA, using a 0.5 mm aluminum filter and an exposure time of 240 ms. Cylindrical crumb samples (30 mm diameter × 35 mm height) were cut out from the center of the cakes and mounted on the scanning platform of the micro-CT. The samples were rotated 180 degrees at a rotation step of 0.3 degrees. The scan time for each sample was approximately 20 min, and the image voxel size was 26.63 μm. The pixel images (1000 × 1000) were then reconstructed and analyzed for porosity using NRecon and CT Analyzer 1.9.1.0 software (SkyScan, Bruker micro-CT, Billerica, MA, USA), respectively.

#### 2.5.5. Specific Volume of Cake

Each cake’s specific volume was determined using the ratio of the cake volume to the weight of the cake. The volume of the cakes was determined using the rapeseed displacement procedure according to AACC 10–05 method [26].

#### 2.5.6. Cake Texture Analysis

The cake texture was measured 24 h after baking using a texture analyzer (TA-Plus texture analyzer, Stable Micro Systems Ltd., Surrey, UK) according to Kim et al. [16] with slight modification. Cake crumbs were cut into 2 cm cubes and compressed two times with a cylindrical probe of diameter 2.5 cm at a speed of 2 mm/s until 50% deformation. The hardness, springiness, cohesiveness, gumminess, and chewiness of the cakes were measured from the texture profile analysis using the Exponent software (version 6.1.4.0, Stable Micro Systems Ltd., Surrey, UK), according to Friedman et al. [27].

### 2.6. Statistical Analysis

All oleogels and oleofoams were prepared in triplicates. Three batches of cakes were prepared for each formulation. All the other experimental measurements were carried out in triplicate using different oleogels, oleofoams, and cake samples, and the average and standard deviation are reported in the manuscript. The results were statistically analyzed from the analysis of variance and *t*-test at a significance level of 5% using Microsoft Excel 2013.

## 3. Results

### 3.1. Formation of Oleofoams from Oleogels

#### 3.1.1. Oleofoam Stability and Overrun

Pea and faba bean protein foam-templated oleogels were prepared by adding canola oil containing either MAG or CW to the freeze-dried foams. Oleogels were also prepared by adding only canola oil, without MAG or CW, to the freeze-dried foam. However, only the oleogels consisting of MAG could be whipped into oleofoam. The air–oil interface stabilization effect of saturated MAG crystals was reported previously by several researchers [17,28,29]. It was proposed that the Pickering-stabilization effect of MAG crystals was responsible for their air bubble stabilization ability [17]. MAGs are surface active, which promotes their adsorption at the air–water interface. Although oleogels consisting of CW also had crystals, these oleogels were completely disrupted and appeared as viscous fluid after whipping. It is possible that the CW crystals could not show the Pickering effect at the canola oil–air interface while whipping due to their inactivity towards surfaces.

Stability and overrun of the oleofoams were determined by the oleogel setting temperature and MAG concentration. The oleofoams prepared from the room temperature-set (25 °C) oleogels displayed much higher stability (Figure 1a) and overrun (Figure 1b) compared to the oleofoams prepared from the refrigerator-set (4 °C) oleogels. After 6 h of storage, oleofoams prepared from room-temperature oleogels displayed 99% stability, while it was only 70% for the oleofoam prepared from refrigerator-temperature-set oleogel (Figure 1a). Overrun of the oleofoams prepared from oleogels stored at room temperature was 157.4 ± 3.7%, which was three times higher than the oleofoam prepared from oleogel at the refrigerator temperature (44.0 ± 1.9%) (Figure 1b). The effect of oleogel set temperature and whipping temperature on oleofoam stability and overrun could be due to the amount, size, and type of MAG crystals formed at different temperatures and on the viscosity and elastic modulus of the oleogels. It was previously reported that the processing conditions significantly vary the MAG crystal properties and alter the foamability and foam stability of the oleogels [28]. The extremely high viscosity of the oleogels formed at freezer temperature might also be why it did not allow the formation of oleofoams. An optimum elastic modulus of the oleogel is essential to obtain a good oleofoam with a high degree of overrun [19]. In the present case, we whipped the oleogels at their respective set temperature; hence, not many air bubbles were incorporated in the highly viscous oleogel set at 4 °C. In contrast, Truong et al. [19] equilibrated a 5 °C set oleogel to room temperature before whipping, leading to a highly stable oleofoam. As expected, the oleogels set in the freezer (−18 °C) did not form any oleofoam when whipping for 45 min due to excessively high viscosity and elasticity of the oleogel.

There was also a significant increase in both stability and overrun of the oleofoams with an increase in MAG concentration (Figure 1c,d). When the MAG concentration was only 0.5%, the stability and overrun after 6 h storage was 30.0 ± 0% and 19.1 ± 2.0%, respectively, which was about 3.3-times and 8-times less than the oleogels consisting of 3% MAG. It should also be noted that at least two to three days of storage was required for 0.5% MAG oleogel to form crystals required for the oleofoam formation. After a day of storage, there were only a few crystals, and while whipping, there was no oleofoam formation. The data shown here were for the oleofoam prepared after 3 days of storage in the case of 0.5% MAG. However, with a higher concentration of MAG, oleofoam could be prepared after a day of storage, and no change in foamability was observed after 3-day storage. When the protein foams were present in the oleogels with 3% MAG, the foam stability and overrun were not significantly different from that of oleogels consisting of only 3% MAG (*p* > 0.05) (Figure 1c,d). Oleofoams prepared from both PPC and FPC protein foam-templated oleogels with 3% MAG displayed 98.0 ± 0% stability after 6 h. After a month of storage, all the oleofoams consisting of 3% MAG (with and without PPC and FPC foam) displayed about 87% stability (Figure 1c). The overrun of the oleofoam prepared from 3% MAG (with and without PPC and FPC foam) was more than 150%, which was significantly higher than that obtained by Gunes et al. [17] and Truong et al. [19], where oleofoams made from 10% MAG oleogel showed an overrun of about 50%. It is possible that with a lower amount of MAG crystals and a lower elasticity of the oleogels, we were able to incorporate more air bubbles in the oleofoams. Heymans et al. [28] also observed a similar high overrun with partially crystalline MAG-containing oleogels.

#### 3.1.2. Viscoelasticity of Oleofoams

An oleofoam used for baking should be able to withstand vigorous mixing conditions used for cake batter preparation. Therefore, it was essential to understand the conditions needed to obtain strong and stable oleofoams from the oleogels. Oscillation strain-dependent viscoelastic measurements (Figure 2a) indicated a gel-like behaviour for all the oleofoams except 1% MAG, where an LVR was observed at a low strain, and the storage moduli (G′) were higher than the loss moduli (G″) within that LVR and below the crossover strain. Similar viscoelastic behaviour of MAG oleofoams was observed previously, and it was reported that the elastic behaviour of the oleofoams might be due to the contributions from both the jamming effect of the MAG crystal-stabilized air bubbles, and the MAG crystal network remained in the oil phase [28]. No LVR was observed for the 1% MAG oleofoam, indicating a weak gel behaviour. The gel strength increased with an increase in concentrations of MAG as well as with the presence of protein foam (Figure 2b). It was reported previously that the oleofoam characteristics were determined by the oleogel characteristics, and when the oleogels can withstand higher stresses before flowing, it results in oleofoams with similar characteristics [28]. The higher gel strength of oleofoams with higher concentrations of MAG might be due to the presence of more MAG crystals at the interface as well as the formation stronger MAG crystal network presented in the continuous phase. The presence of a protein foam network in the oil phase of the foam-templated oleogels also improved gelation in the oleofoams (Figure 2b). Interestingly, when a conventional shortening was whipped to create oleofoam (after similar room temperature storage), it showed the highest gel strength of oleofoams developed in the present work (Figure 1b), which could be due to the presence of both saturated triacylglycerols and mono-diglycerols present in the shortening.

The time-dependent viscoelastic behaviour of the oleofoams was determined using frequency sweep measurements (Figure 2c). All the oleofoams displayed higher G′ than G″ over the entire frequency range studied, indicating domination of the elastic behaviour over the viscous. It should be noted that although the 1% MAG oleofoam did not show any LVR (Figure 2a), the frequency sweep at 0.05% strain still showed a dominant elastic behaviour although the gel was weak. A similar range of elasticity was also observed by Heymans et al. [28] in their MAG-based oleofoams. All the oleofoams displayed similar changes in G′ and G″ with frequency; G′ increased with frequency, and G″ showed downward peaks, indicating minor rearrangements within the foam structure during the measurement. This structural rearrangement and increase in gel strength might be due to oil drainage at a higher frequency. However, the slope of the G′ curve was only slightly positive, indicating limited rearrangement of the gel structure as a function of frequency. Both G′ and G″ of shortening oleofoams were higher than that of MAG-stabilized oleofoams, which might be due to the larger amount of saturated crystalline fat in the shortening than the MAG-stabilized oleogels. The viscoelastic moduli of the oleofoams increased with MAG concentration, while no significant difference was found between the oleofoams stabilized by 3% MAG and 3% MAG with protein foam-templates (*p* > 0.05). The tan δ (ratio of G″ to G′) values (Figure 2d) for all oleofoams decreased with an increase in frequency and then increased again. Still, the values were less than 1 throughout the frequency range studied, indicating dominant elastic behaviour. The frequency at which tan δ reversed increased to a higher value with an increase in MAG concentration or due to an increase in the crystallinity in shortening. It indicates a shift in the oleofoam structure from increased elastic behaviour toward a reduction in elasticity as a function of frequency. Oleofoams with a higher elasticity needed a higher frequency to induce such a change in rheological behaviour.

#### 3.1.3. Visual Observation and Microstructure of Oleofoams

Visually, the shortening oleofoam appeared the fluffiest and had the best ability to hold its peak (see Figure 3 visual observation). The MAG-only oleofoam appeared the softest of all (Figure 3). The MAG plus pea or faba protein foam-templated oleofoams were able to hold their peaks better than the MAG-only oleofoam, which matched with elastic modulus in the LVR (Figure 2b). A clear difference in oleofoam microstructure can be observed in Figure 3. The shortening oleofoam showed the smallest air bubbles of all and a jammed structure, which was responsible for their highest elastic modulus. The MAG-only oleofoam showed the largest bubble size. We used only 3% MAG to form the oleogel in contrast to the 10% MAG used by other researchers [17,19], which could be insufficient to stabilize many smaller air bubbles. The oleofoam developed from MAG plus protein foam-templated oleogels showed smaller and more packed air bubbles than MAG-only oleofoam. Such improvement in the oleofoam microstructure could be due to the protein’s ability to stabilize air bubbles and provide a network in the continuous oil phase to restrict air bubble movement. Note that the initial oleogels were formed by adsorbing canola oil in the freeze-dried protein foams, where the oil was adsorbed in the hydrophobic air pockets of the denatured protein foam network [6]. The protein network in the oil phase of the oleogel helped provide improved stability of the oleofoam.

### 3.2. Cake Baking with Oleogels and Oleofoams

Shortening is added to bakery products to stabilize air bubbles in the batter, shorten the gluten network, and provide lubrication, which helps form soft and tender baked products [30]. To investigate the effectiveness of oleofoams in mimicking these characteristics, different oleofoams were used to completely replace shortening in cake batter preparation. Since a minimum of 3% MAG in oleogel was required to form stable oleofoams, various oleofoams prepared from 3% MAG, 3% MAG with PPC foam, and 3% MAG with FPC foam-templated oleogels were used for cake batter preparation. It was hypothesized that the incorporation of air in the batter would be increased due to the presence of oleofoams resulting in high-quality cakes. Shortening was also completely replaced with canola oil as a control. (The oleofoams contained more than 95% liquid canola oil.) Since the oleogels consisting of CW could not make any oleofoams, they were excluded from the cake batter preparation.

#### 3.2.1. Optimization of Cake Batter Preparation Using Oleofoams

The specific gravity of the cake batters prepared using shortening and 3% MAG oleofoam using the AACC method is shown in Table 1. Specific gravity indicates the amount of air incorporation in the batter, and the lower the value, the higher the air incorporation. The specific gravity of the 3% MAG oleofoam batter was 1.17 ± 0.002, which was significantly higher than that of the shortening batter, 0.88 ± 0.02 (*p* < 0.05) (Table 1). Specific gravities of 3% MAG oleofoam and 3% MAG oleogel batters were not significantly different (*p* < 0.05), which indicates that the air bubbles generated through oleofoams were destroyed during the cake batter preparation using the AACC method. The confocal microstructure of the cake batters also supports this hypothesis (Figure 4, discussed below). Therefore, a gentler mixing procedure was needed while using the oleofoams for cake batter preparation. However, it was also necessary to ensure proper mixing of all other ingredients in the batter, which was essential to produce high-quality cakes. The batter prepared using the new lab-developed procedure (described in Section 2.4) showed a specific gravity of 1.04 ± 0.01 for the 3% MAG oleofoam, which was significantly lower (*p* < 0.05) than the batter made using the same oleofoam using the original AACC method (1.17 ± 0.002, Table 1). Therefore, this new method was adopted for the cake batter preparation using oleofoams.

#### 3.2.2. Microstructure of Cake Batters

The confocal microscopy of cake batter prepared with shortening using the AACC method showed numerous air bubbles (appeared as dark circles) in the fat phase (appeared in red) (Figure 4a,h), which were stabilized by crystalline saturated triglycerides and mono-diglycerides of shortening [31]. There was a clear separation between the fat phase and the protein–carbohydrate-rich phase (green) in the shortening batters prepared using the AACC method (Figure 4a,h) or the new method (Figure 4d,k). However, no such clear boundary between the fat phase (red) and the protein–carbohydrate-rich phase (green) was observed in the batters produced with the MAG oleofoam AACC method (Figure 4b,i) and canola oil using the new method (Figure 4c,j). The lack of air bubbles in the fat phase of the MAG oleofoam batter prepared using the AACC method could be due to the disruption of MAG oleofoams during the batter preparation process. For the canola oil batter, it could be due to the inability of canola oil to stabilize air bubbles; therefore, even with the new method, it was not possible to incorporate more air bubbles in the canola oil cake batter. The microstructure of the canola oil batter in the present study was similar to that of cake batter prepared using rapeseed oil [32], which was ascribed to the inability of the liquid oil to stabilize air bubbles and poor gluten development. A decrease in specific gravity of 3% MAG oleofoam cake batter made using the new method compared to the conventional AACC method might be due to more air incorporated via oleofoam, which can be seen from their microstructures (Figure 4e–n). In the oleofoam batters prepared using the new method, a clear distinction between the air-bubble-rich fat phase and the protein–carbohydrate phase was observed (Figure 4e–n), which was somewhat similar to that of the shortening batter. However, the air bubbles in the fat phase of the oleofoam batters prepared using the new method were lower in number and larger in size compared to the shortening batter prepared using the AACC method, which could be responsible for their higher specific gravity reported in Table 1.

Resembling what was observed in the confocal micrographs, the X-ray microtomography images of cake batters (Figure 5) also support more air incorporation in the MAG oleofoam batters prepared using the new method than the one prepared using the AACC method. To quantify the air incorporation, the total porosity values of the batters were calculated from the microtomography images (Figure 6a). Both the shortening batters produced using the AACC method and the shortening oleofoam batters produced using the new method displayed significantly higher porosity than the other batters (Figure 6a) (*p* < 0.05). A higher number of open pores in the shortening batter compared to oleogel and liquid oil batters were observed previously by Kim et al. [16]. As expected, the MAG oleofoam batter produced using the AACC method displayed the least porosity, followed by the canola oil batter. The three oleofoam batters made with MAG, MAG+PPC foam, and MAG+FPC foam using the new method showed significantly higher porosity than the MAG oleofoam AACC and canola oil batter (Figure 6a), which agreed with our initial hypothesis that incorporation of oleofoam would increase the incorporation of air in the batter provided a gentler mixing process was used during the batter-preparation process.

#### 3.2.3. Rheology of Cake Batters

The rheology of all batters is shown in Figure 7. All batters except MAG-AACC and canola oil batter displayed a brief linear viscoelastic region at the low-strain region followed by a drop in G′ and crossover of G′ and G″ at around 10% strain (Figure 7a,b). To compare the gel strength of the batters, their moduli at 0.1% strain (within the LVR) were plotted in Figure 7c. All batters showed G′ > G″, indicating gelation. The shortening batter prepared with either the AACC or the new method displayed the highest G′ of all batters, followed by MAG+PPC foam, MAG+FPC foam, and the MAG-only batters prepared with the new method. The lowest elastic modulus was observed with the MAG-AACC and canola oil batters, possibly due to the lack of air incorporation. In the frequency-sweep study, all batters displayed higher G′ than G″ in the whole frequency range studied (Figure 7d). Both G′ and G″ also increased linearly with an increase in frequency in a log-log scale, indicating weak gel behaviour. The batters prepared with shortening using the AACC and the new method displayed the highest and most consistent G′ across the frequency range studied. The canola oil batter, although it showed a very low G′ at a lower frequency, increased with an increase in frequency and reached a value near the shortening-AACC batter, which could be due to compaction in structure at a higher frequency due to the lack of air bubbles. A similar change in elastic moduli with frequency was also observed for the MAG-AACC batters (Figure 7d). The batters prepared with the new method consisting of MAG oleofoam and MAG+protein oleofoams displayed a similar consistent change in moduli as a function of frequency as with the shortening batters. The tan δ values of all batters were less than 1 in the whole frequency range (Figure 7e). With an increase in frequency, the tan δ decreased to a minimum, indicating strengthening of gel structure followed by an increase due to structural weakening. The shortening batters displayed the lowest tan δ values, indicating that the replacement of shortening with MAG oleofoams led to a greater contribution to the viscous nature of the batter.

The viscosity of all batters decreased with an increase in shear rate, indicating a pseudoplastic behaviour (Figure 7f). The initial increase in viscosity for some samples observed at a very low shear rate could be due to an experimental artifact. An onset of Newtonian plateau was also observed for most samples at both low and high shear rates. Nevertheless, a simple comparison among all the samples was done using a power-law model ($ɳ=K{\dot{\left(\gamma \right)}}^{n-1}$ (where ɳ is the viscosity, $\dot{\gamma }$ is the shear rate, *K* is the consistency coefficient, and *n* is the flow behaviour index) was used to fit the flow curves of all cake batters. The fitting parameters are shown in Table 2. R^2^ values were >0.96 for all except for the canola oil batters (R^2^ 0.87). Flow behaviour indices of all batters were less than 1, and shortening batter prepared using the AACC method displayed the lowest flow behaviour index, indicating the most shear-thinning due to breakdown of higher-ordered structure, followed by all the MAG oleofoam batters (*p* < 0.05). Canola oil batter displayed the highest flow behaviour index, indicating the least shear thinning and the formation of a less ordered structure. The shortening batter obtained using the AACC method displayed the highest consistency index (K), followed by the shortening oleofoam batter obtained using the new method (Table 2), indicating the highest viscosities at a shear rate of 1 s^−1^ compared to all others. The low-shear viscosity of the batters was also compared using the values at 0.015 s^−1^ (Appendix A). The two shortening batters also showed the highest low-shear viscosities, which might be attributed to the solid fat crystals and higher air incorporation in the batter. The viscosities of the shortening batters, however, became close to the viscosities of all other batters at higher shear rates (Figure 7f) due to the disruption of the fat crystal network in the shortening and the collapse of the air bubbles. Including a higher portion of liquid oil in the oleofoam and canola oil batters led to lower viscosities at a lower shear rate than the shortening batter. In fact, the lowest low-shear viscosity of all batters was observed for the canola oil batter (Appendix A). Similar viscosity reduction was observed previously when shortening was replaced with rapeseed oil [32]. All three oleofoam batters containing MAG or MAG+protein foams and the MAG-AACC better showed similar but lower consistency coefficients (Table 1) and low-shear viscosities (Appendix A) than the shortening batters, which could be due to a reduced fat crystal network in the oleofoams compared to the shortening and partial collapse of the air bubbles of the oleofoams. Interestingly canola oil batters showed higher consistency coefficients (viscosity at a shear rate of 1 s^−1^, Table 2) than the MAG+protein foam-based oleofoam batters, which indicates that at 1 s^−1^ shear rate, the structural breakdown of the MAG+protein foam-based oleofoam batters was even higher than the canola oil batters.

### 3.3. Properties of Cakes

The cross-sectional view of the cakes is provided in Figure 8. Both the shortening cakes prepared using the AACC method (Figure 8a) and shortening oleofoam cake prepared using the new method (Figure 8d) displayed a uniform distribution of small and open-air cells and small granular particles of flour solids. On the other hand, the other oleofoam cakes displayed a non-uniform distribution of large air bubbles. Microtomography images showed that the pores in the oleofoam cakes (Figure 9e–g) were larger and non-homogeneously dispersed compared to the shortening cakes (Figure 9a,d). It indicates that the air bubbles were merged during the expansion of the batter in the baking process, which might be due to the poor rheological properties of the batters (Figure 7) due to the incorporation of liquid oil in oleofoams. The air bubbles of the oleofoams in the batter might not have transferred properly into the protein-rich phase of the cake during the baking process, as in the case of the shortening batter. Although the total porosity of the oleofoam cakes (Figure 6b) prepared using the new method was similar to that of shortening cakes, the pores were much larger with the oleofoam cakes made with the new method (Figure 9). Nevertheless, the total porosity of the three MAG and protein foam-stabilized oleofoam cakes prepared using the new method was still significantly higher than the MAG AACC method (Figure 6b), which indicates an improvement in cake property due to the use of a gentler mixing process during batter preparation.

Interestingly, the cake prepared using canola oil using the new method displayed a close similarity in microstructure (Figure 9) and total porosity (Figure 6b) with shortening cakes although their values in the batter stage (Figure 6a) were quite different. Therefore, the stepwise addition of liquid canola oil during batter preparation helped distribute the air bubbles generated during the baking process. It is well-known that the air bubbles generated during the batter preparation expand during baking due to the presence of leavening agents. Above the fat-melting temperature during baking, the air bubbles in the fat phase migrate to the protein-rich phase [30]. The ability of cake batters to keep the air bubbles in the batter and at the same time to allow the cake to rise is determined by the rheology of the cake batter [33]. The specific volume of the cake (Table 1) showed that, although the oleofoam cakes made with the new method showed significantly higher cake volume than the oleofoam-AACC cake, their values were still lower than the shortening-AACC cakes. The volume of the cake therefore not only determined by the air incorporated in the batter but the ability of the batter to keep the air bubbles from coalescing during baking. This shows that the viscosity and gel strength of the cake batter made with the oleofoams were not high enough, or the cake crumb structure was not firm enough to keep the air bubbles in the oleofoam batters and cakes from coalescing during baking despite using the new gentler batter-preparation method.

Textural properties of cakes such as hardness, springiness, cohesiveness, and chewiness are displayed in Figure 10. Hardness is an indicator of staling of baked products and can be determined using the force required to compress the sample to a certain height. Springiness is an indication of the elasticity of the sample. Cohesiveness is a direct function of work required to break down the internal bonds between different ingredients in the cake matrix during each “chew” or the internal resistance of food to traction, which was determined from the ratio of the second to the first peak area of the two-bite test. Finally, chewiness values indicate the energy required for the mastication of cakes, which was calculated from the product of the primary parameters: hardness, cohesiveness, and elasticity [27]. All the oleofoam and canola oil cakes prepared using the new method displayed similar hardness and chewiness as that of shortening cake prepared using the AACC method (*p* > 0.05) (Figure 10a), which indicates that the stepwise addition of the fat phase to the batter helps to form a cake with higher textural quality. The cakes prepared using the 3% MAG oleofoam using the AACC method displayed the highest hardness and chewiness compared to all other cakes (Figure 10a,d), indicating that the AACC method is not ideal for the new oleofoam developed in this research. The springiness of the new method-based oleofoams cakes was, however, higher than that of shortening cake prepared using the AACC method, indicating that once the deformation happened to the oleofoam cakes, it could recover the structure well compared to that of shortening cake produced using the AACC method.

Overall, the textural properties of the cakes were related to their microscopic properties. When more open pores and softer gluten-gelatinized starch networks formed in the cakes, they displayed lower hardness and chewiness, as can be seen in the MAG oleofoam new-method-based cake compared to the MAG oleofoam-AACC cake. Although shortening-AACC and the new-method-based MAG and MAG+protein foam-stabilized oleofoam cakes displayed similar porosity (Figure 6b), the voids presented in the oleofoam cakes were larger and isolated (Figure 9e–g). During deformation, these air bubbles could maintain their structure and partially recover when the deformative force was removed, leading to a higher springiness than the shortening-AACC cake. In our previous study, the MAG+pulse protein foam-stabilized oleogel produced a harder cake than the shortening cake [15]. Compared to that, the oleofoams (developed from the oleogels) produced a softer cake, similar to the shortening cake, when the new lab-developed batter-preparation process was adopted. However, further research would be needed to keep the oleofoams stronger and more stable during the baking process to ensure a smaller air bubble size for the long-term storage quality of cakes.

## 4. Conclusions

Pulse protein foam-templated oleogels consisting of MAG can be converted into highly stable oleofoams by whipping the oleogels set at room temperature. The presence of MAG was essential to form air bubbles in the oleofoams, where the overrun increased with MAG concentration. Oleofoams with the highest stability and overrun were obtained when the oleogels consisted of 3% MAG with the pulse protein foam. The oleofoam developed from MAG+protein foam-templated oleogels showed smaller and more packed air bubbles than MAG-only oleofoam, which could be due to the protein’s ability to stabilize air bubbles and also provide a network in the continuous oil phase to restrict air bubble movement. It was hypothesized that the incorporation of the oleofoams in the cake batter would increase air incorporation in the batter, which in turn would improve the textural qualities of the cakes compared to the cake made with oleogels. A novel gentler mixing method involving multistep oleofoam addition for batter preparation was found to be more effective in retaining the oleofoam structure in the batter with a uniform air bubble distribution and higher total porosity than that obtained using the AACC method. The X-ray microtomography images of the cakes showed a non-homogeneous distribution of much larger air bubbles in the new method-based oleofoam cakes compared to the shortening cakes although the total porosity values were not significantly different. The textural properties of the cakes showed that oleofoam cakes made with the new method yielded similar hardness and chewiness compared to the shortening cakes, along with slightly higher springiness values. Therefore, by optimizing the methodology to increase air incorporation and improving batter rheology, high-quality cakes can be obtained with oleofoams made from pulse protein foam-templated oleogels as a shortening alternative. The findings from this work showed a value-added utilization of plant proteins in oil structuring to create a novel fat substitute; however, further economic feasibility analysis must be considered before it can be utilized commercially.

## Figures and Tables

**Figure 1 foods-11-02887-f001:**
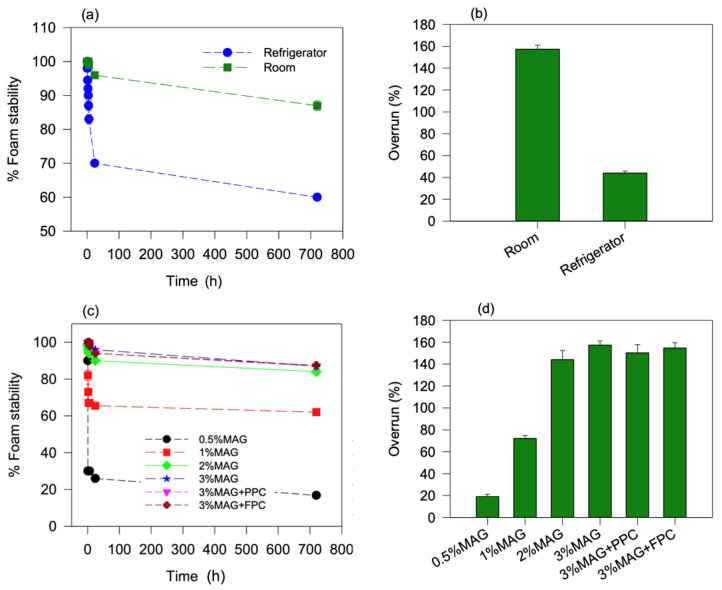
(**a**) Foam stability and (**b**) overrun of oleofoams obtained using 3% MAG oleogels set at different temperatures. (**c**) Foam stability and (**d**) overrun of oleofoams obtained from oleogels set at room temperature with different concentrations of MAG and PPC and FPC protein foam-templated oleogels with 3%MAG. MAG, monoacylglycerols; PPC, pea protein concentrate-stabilized foam; FPC, faba bean protein concentrate-stabilized foam.

**Figure 2 foods-11-02887-f002:**
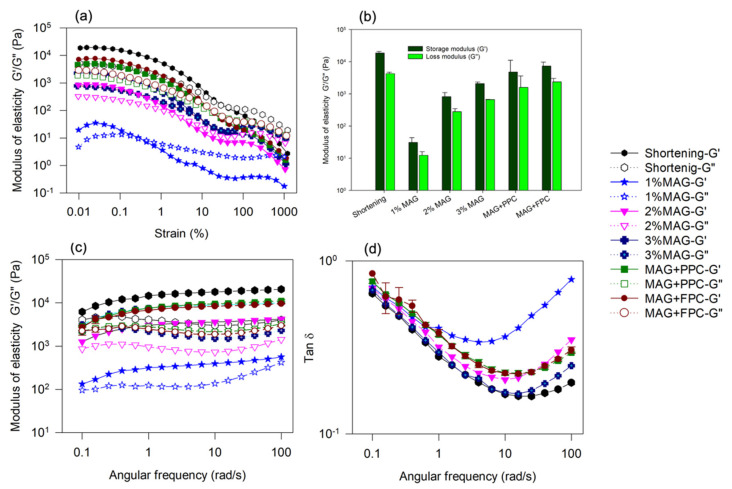
(**a**) Viscoelastic modulus of oleofoams measured as a function of oscillation strain% at a constant frequency of 0.5 Hz at room temperature. (**b**) Comparison of storage (G′) and loss moduli (G′) of the oleofoams at 0.05% strain. (**c**) Viscoelastic moduli and (**d**) tan δ of oleofoams as a function of angular frequency at a constant oscillation strain of 0.05%. MAG, monoacylglycerols; PPC, pea protein concentrate-stabilized foam; FPC, faba bean protein concentrate-stabilized foam.

**Figure 3 foods-11-02887-f003:**
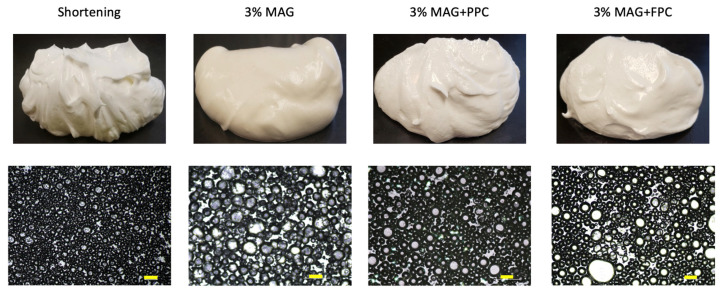
Visual observation and microstructure of different oleofoams prepared by whipping shortening and various MAG-based and MAG plus protein foam-templated oleogels. The scale bar indicates 100 mm. MAG, monoacylglycerols; PPC, pea protein concentrate-stabilized foam; FPC, faba bean protein concentrate-stabilized foam.

**Figure 4 foods-11-02887-f004:**
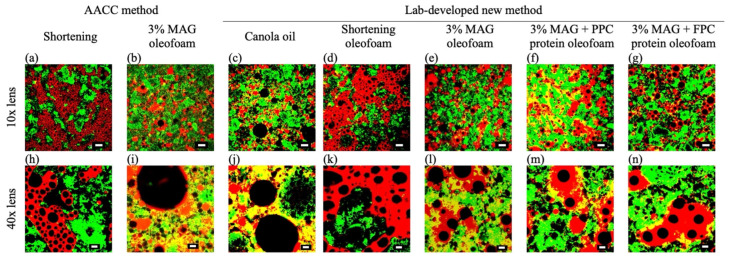
The confocal microscopic images of cake batters prepared using various fats using either AACC or lab-developed method. AACC method: Shortening (**a**,**h**), MAG oleofoam (**b**,**i**). Lab-developed method: canola oil (**c**,**j**), shortening oleofoam (**d**,**k**), 3%MAG oleofoam (**e**,**l**), 3% MAG with PPC protein foam-templated oleofoam (**f**,**m**), and 3%MAG with FPC protein foam-templated oleofoam (**g**,**n**). Images were taken using 10X (top row, **a**–**g**, scale bar 100 µm) and 40X (bottom row, **h**–**n**, scale bar 20 µm) objective lens. The oil phase in the cake batters was stained using Nile red (red color), and the protein was stained with fast green (green color). The dark phase indicates starch and water, and the dark bubbles indicate air bubbles. MAG, monoacylglycerols; PPC, pea protein concentrate-stabilized foam; FPC, faba bean protein concentrate-stabilized foam.

**Figure 5 foods-11-02887-f005:**
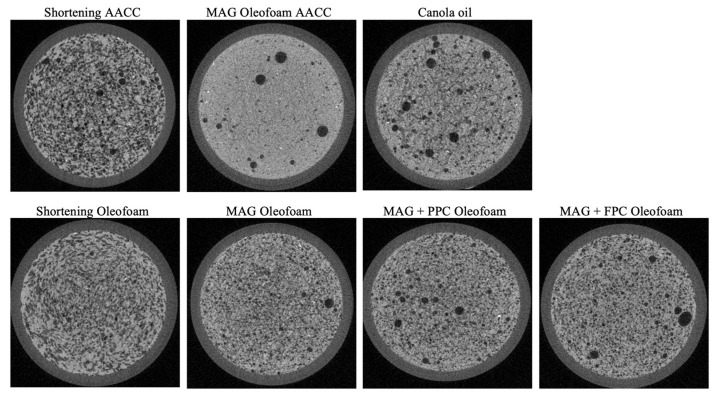
X-ray microtomography images of different cake batters. The names of the batters are shown on the top of each image. Shortening AACC and MAG oleofoam AACC batters were prepared using the AACC method, and the rest of the batters were prepared using the new lab-developed method. MAG, monoacylglycerols; PPC, pea protein concentrate-stabilized foam; FPC, faba bean protein concentrate-stabilized foam.

**Figure 6 foods-11-02887-f006:**
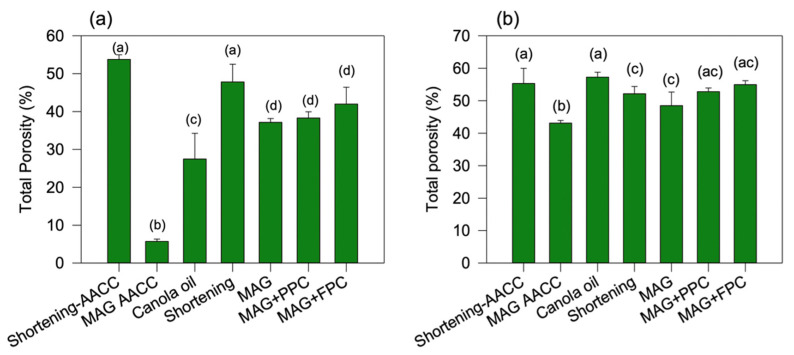
Total porosity of (**a**) batters and (**b**) cakes calculated from X-ray microtomography images. The batters and cakes were prepared with shortening and 3% MAG oleofoam using the AACC method (shortening-AACC and MAG-AACC, respectively). Batters and cakes prepared with canola oil, shortening oleofoam, 3%MAG oleofoam, 3%MAG+PPC oleofoam, and 3%MAG+FPC oleofoam using the new method developed in the lab (shortening, MAG, MAG+PPC, and MAG+FPC, respectively). Different letters in each figure indicate a statistically significant difference (*p* < 0.05). MAG, monoacylglycerols; PPC, pea protein concentrate-stabilized foam; FPC, faba bean protein concentrate-stabilized foam.

**Figure 7 foods-11-02887-f007:**
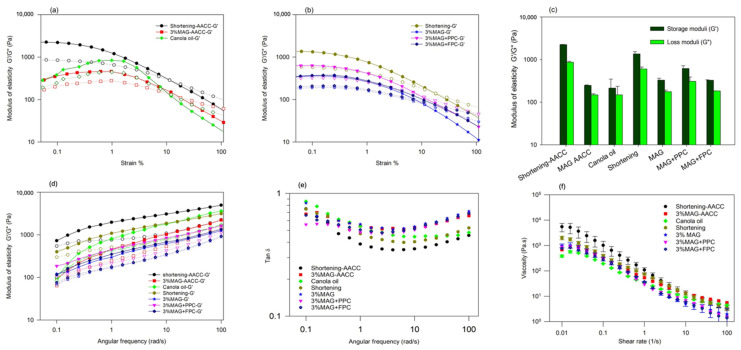
Rheology of different cake batters: (**a**,**b**) Elastic moduli as a function of strain and (**c**) storage (G′) and loss (G″) moduli at a constant strain of 0.1%. (**d**) Elastic moduli and (**e**) tan delta as a function of frequency. (**f**) Viscosity as a function of shear rate. G′ values are shown with filled symbols, while the G″ are represented with open symbols. Shortening-AACC and 3%MAG AACC batters were produced using the AACC method. All the other batters, prepared with canola oil or oleofoams, used the new method developed in the lab. MAG, monoacylglycerols; PPC, pea protein concentrate-stabilized foam; FPC, faba bean protein concentrate-stabilized foam.

**Figure 8 foods-11-02887-f008:**
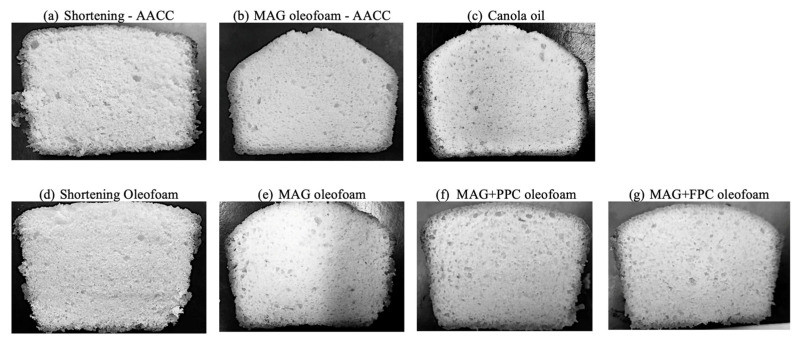
Cross-sectional view of different cakes produced using various fats with either AACC or lab-developed new method. AACC method: (**a**) shortening and (**b**) MAG oleofoam. Lab-developed method: (**c**) canola oil, (**d**) shortening oleofoam, (**e**) 3%MAG oleofoam, (**f**) 3%MAG+PPC oleofoam, and (**g**) 3%MAG+FPC oleofoam. MAG, monoacylglycerols; PPC, pea protein concentrate-stabilized foam; FPC, faba bean protein concentrate-stabilized foam.

**Figure 9 foods-11-02887-f009:**
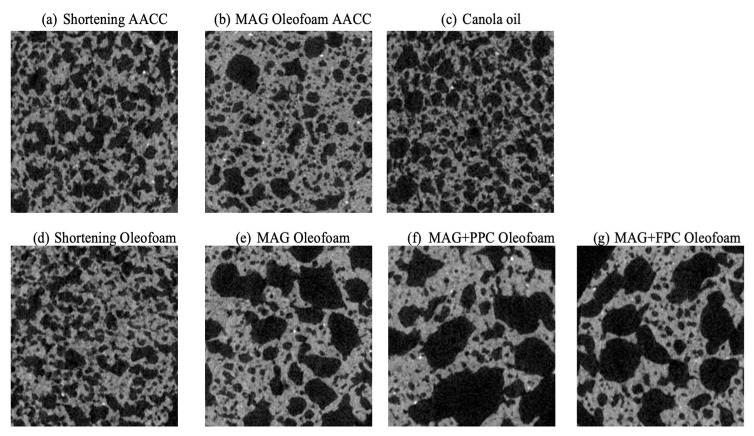
Microtomography images of different cakes. Names of cakes are shown on the top of each image. (**a**) Shortening AACC and (**b**) MAG oleofoam AACC cakes were prepared using the AACC method, and (**c**–**g**) the rest of the cakes were prepared using the new lab-developed batter preparation method. MAG, monoacylglycerols; PPC, pea protein concentrate-stabilized foam; FPC, faba bean protein concentrate-stabilized foam.

**Figure 10 foods-11-02887-f010:**
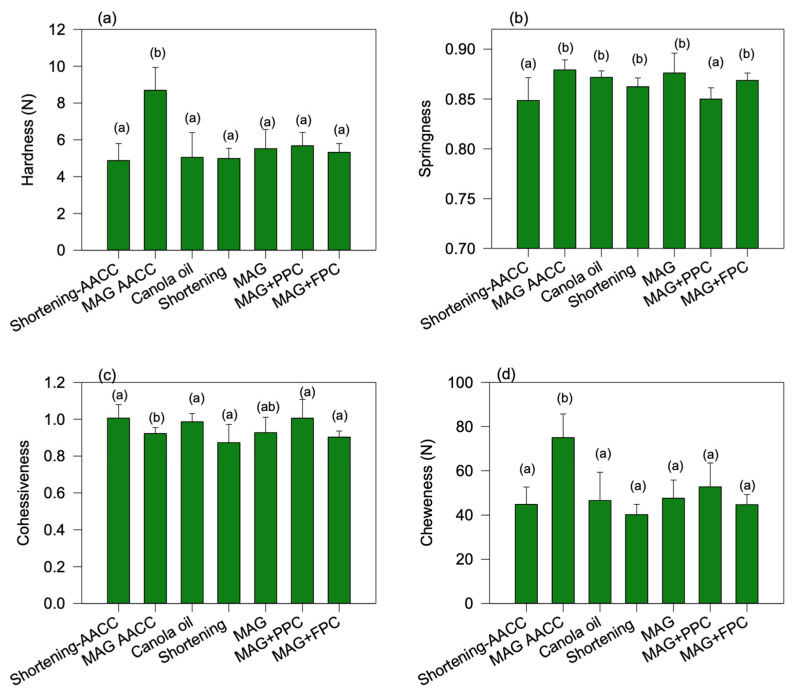
Texture analysis of different cakes measured 24 h after baking at room temperature: (**a**) hardness, (**b**) springiness, (**c**) cohesiveness, and (**d**) chewiness of cakes prepared using various fats with either AACC or the new lab-developed method. AACC method: Shortening-AACC, MAG oleofoam-AACC. The new lab-developed method: canola oil, shortening oleofoam, 3%MAG oleofoam, 3%MAG+PPC oleofoam, and 3%MAG+FPC oleofoam. MAG, monoacylglycerols; PPC, pea protein concentrate-stabilized foam; FPC, faba bean protein concentrate-stabilized foam.

**Table 1 foods-11-02887-t001:** The specific gravity of different cake batters and specific volume of corresponding cakes made with shortening and MAG-oleofoam using the AACC method and a lab-developed method. Different letters in each column indicate statistically significant differences among the mean (*p* < 0.05).

	Specific Gravity of Cake Batter	Specific Volume of Cake (cm^3^/g)
Shortening-AACC	0.88 ± 0.02 ^a^	2.21 ± 0.06 ^a^
MAG oleofoam-AACC	1.17 ± 0.00 ^b^	1.58 ± 0.02 ^b^
Canola oil	1.19 ± 0.00 ^c^	1.99 ± 0.02 ^c^
Shortening Oleofoam	1.01 ± 0.02 ^d^	1.74 ± 0.05 ^d^
MAG Oleofoam	1.04 ± 0.01 ^d^	1.73 ± 0.03 ^d^
MAG+PPC Oleofoam	1.05 ± 0.03 ^d^	1.82 ± 0.03 ^e^
MAG+FPC Oleofoam	1.03 ± 0.03 ^d^	1.79 ± 0.05 ^ed^

**Table 2 foods-11-02887-t002:** Effect of replacement of shortening by different oleofoams on power-law parameters of different cake batters. The batters were prepared with shortening or MAG-oleofoam using the AACC method or with different oleofoams using a lab-developed method. Different letters in each column indicate a statistically significant difference (*p* < 0.05).

Sample	Flow Behaviour Index (n)	Consistency Coefficient (K, Pa·s^n^)	R^2^
Shortening-AACC	0.07 ± 0.01 ^a^	118.47 ± 2.19 ^a^	0.9998
MAG oleofoam-AACC	0.19 ± 0.02 ^b^	56.29 ± 2.07 ^b^	0.9984
Canola Oil	0.55 ± 0.06 ^c^	70.78 ± 17.07 ^c^	0.8737
Shortening oleofoam	0.33 ± 0.03 ^d^	97.07 ± 10.09 ^d^	0.9878
MAG Oleofoam	0.19 ± 0.13 ^b^	45.19 ± 1.67 ^e^	0.9620
PPC+MAG Oleofoam	0.19 ± 0.02 ^b^	42.60 ± 15.16 ^b,e^	0.9974
FPC+MAG Oleofoam	0.25 ± 0.16 ^b,d^	46.03 ± 2.13 ^e^	0.9974

## Data Availability

The data presented in this study are available on request from the corresponding author.

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
