# Peer review of "Conversion of Pulse Protein Foam-Templated Oleogels into Oleofoams for Improved Baking Application"

_foods, 2022, doi:10.3390/foods11182887_

Round 1
Reviewer 1 Report
This study explored the use of oleofoams obtained by whipping the pulse protein foam-templated oleogel for cake baking. The article is well written and discussed and is of high relevance to the area of oil structuring and its application. Yet some methodological procedures and discussion are still unclear, therefore I recommend some further clarification is carried out to improve the quality and understandability of the manuscript.
Abstract: Include here that oleogels/oleofoam were also produced using CW and that it was not possible to produce oleofoams with CW and for that reason it was no incorporated in cake
It is a bit confusing why the beakers were then quickly transferred to a refrigerator (4 ºC), or a freezer (-20 ºC) or left at room temperature (22 ºC) to allow formation of oleogel? Were these 3 approaches compared? Moreover, the oil was simply poured at the freeze-dried foam. The was no shearing step? Many studies report that for the formation of oleogel by indirect approaches needs to be sheared. It seems to me that this study is a mixture of direct and indirect method, is this the case? Please clarify this and why there was no shearing step.
It is a bit confusing how overrun was calculated as authors mentioned 200 g of oleogel was whipped to form oleofoam, but in equation 1 it reads 50mL of oleogel. Shouldn’t overrun be calculated over the entire produced volume? Or did author measure the weight of 50mL in the same cylinder? Please clarify.
Please clarify what was meant by to drain liquid oil in Line 153? Did the oil separate from oleofoams? It would be nice to present some pictures of the appearance of the oleofoam immediately after production and after 6h to visually the oil drainage.
Why were cake baked “until done”? Shouldn’t the baking time be a fixed parameter? Longer cooking times leads to more liquid evaporation and therefore harder texture. Without a fixed baking time one cannot know if the difference in texture is due to the different fat types or the baking time. Therefore, I have doubts the entire discussion of texture is valid as texture can be directly related to cooking time.
In section 3.1.1. authors claim a Pickering-stabilization effect; however, no analysis was carried out to confirm that the oleogel had this effect. This discussion was based on another study with MAG
In the discussion/conclusion some points should be further discussed/mentioned: Are the properties of MAG + protein foam-templated oleogels considerable better than solely oil or MAG to justify the costs involved in the lyophilization step required for the production of foam-templated oleogels? Also, when considering feasibility and industrial scaling up, is it justified to include a process in which one must first freeze samples (1 day), lyofilize (2 day), incorporate oil and rest for 1 day? This includes at least 4 days just to produce fat. Please share these thoughts in your conclusion taking into consideration other oil structuring approaches.
Other minor comments:
Line 35: Merge references
Line 39: English check
Line 98: Simulated? Simulated in terms of what?
Line 128: Better write oils containing CW and MAG…
Line 131: remove “was”
Line 142: Add rpm of the mixer as was done in section 2.2
Line 165 and 168: Why after 30 or 5 min? Please clarify
Line 181? Why so much sugar?
Line 192: provide rpm
Line 211: Please provide reference for this methodology. Why not measure Apparent density instead?
Line 288: Why was canola oil added to the shortening?
Line 383: Which peak?
Author Response
This study explored the use of oleofoams obtained by whipping the pulse protein foam-templated oleogel for cake baking. The article is well written and discussed and is of high relevance to the area of oil structuring and its application. Yet some methodological procedures and discussion are still unclear, therefore I recommend some further clarification is carried out to improve the quality and understandability of the manuscript.
Authors’ response: Thanks for your time in reviewing our manuscript. We have taken great care to address your comments to improve the quality of the manuscript. All the changes are highlighted in the revised version.
Abstract: Include here that oleogels/oleofoam were also produced using CW and that it was not possible to produce oleofoams with CW and for that reason it was no incorporated in cake
Authors’ response: It is now included in the abstract. Line 16.
It is a bit confusing why the beakers were then quickly transferred to a refrigerator (4 ºC), or a freezer (-20 ºC) or left at room temperature (22 ºC) to allow formation of oleogel? Were these 3 approaches compared?
Authors’ response: Yes, our goal to find out which of these oleogel gives the best oleofoam. This was discussed in detail in the manuscript section 3.1.1 and Figure 1a and 1b. In lines 305-311, we have also discussed why the freezer-set oleogel was no good.
Moreover, the oil was simply poured at the freeze-dried foam. The was no shearing step? Many studies report that for the formation of oleogel by indirect approaches needs to be sheared. It seems to me that this study is a mixture of direct and indirect method, is this the case? Please clarify this and why there was no shearing step.
Authors’ response: yes, the oil was poured at the F/D foam. We found that shearing would damage the foam structure. This has been discussed in our previous papers and we have given those references in the methodology section 2.2. We have also clarified this in the manuscript. See lines 135-6.
It is a bit confusing how overrun was calculated as authors mentioned 200 g of oleogel was whipped to form oleofoam, but in equation 1 it reads 50mL of oleogel. Shouldn’t overrun be calculated over the entire produced volume? Or did author measure the weight of 50mL in the same cylinder? Please clarify.
Authors’ response: We have measured weight of 50 ml oleofoam in a separate glass graduated measuring cylinder. This was mentioned in the methodology (section 2.3.1). We have also made it clear that a part of the oleofoam was used for this. See line 150.
Please clarify what was meant by to drain liquid oil in Line 153? Did the oil separate from oleofoams? It would be nice to present some pictures of the appearance of the oleofoam immediately after production and after 6h to visually the oil drainage.
Authors’ response: When one measures foam stability or liquid drainage for an aqueous foam, they measure the liquid aq. phase drainage at the bottom of the foam. Similarly, here we measured the liquid oil drainage at the bottom of the oleofoam. Its is simply a foam at the top and liquid oil at the bottom. To keep things simple, we are not adding any photos of this.
Why were cake baked “until done”? Shouldn’t the baking time be a fixed parameter? Longer cooking times leads to more liquid evaporation and therefore harder texture. Without a fixed baking time one cannot know if the difference in texture is due to the different fat types or the baking time. Therefore, I have doubts the entire discussion of texture is valid as texture can be directly related to cooking time.
Authors’ response: This was done based on AACC method. If we fix baking time, then not all cakes would be baked properly and there will be chance of less-baked cakes. “Until done” means inserting a toothpick in the cake and seeing whether the crust of sticking to it. Although we said until done, baking for all cakes were varied between 20-25 minutes based on formulations. We have now added this in the manuscript. See line 213-4.
In section 3.1.1. authors claim a Pickering-stabilization effect; however, no analysis was carried out to confirm that the oleogel had this effect. This discussion was based on another study with MAG
Authors’ response: Pickering effect was not for the oleogel, both MAG and CW formed oleogels. But only MAG-oleogels were able to withstand whipping and formed oleofoam. MAGs are known to be surface active, and their Pickering stabilization ability is well documented. We have given three references to support this in section 3.1.1 (line 284). The text has also been modified to clarify (see line 285-6).
In the discussion/conclusion some points should be further discussed/mentioned: Are the properties of MAG + protein foam-templated oleogels considerable better than solely oil or MAG to justify the costs involved in the lyophilization step required for the production of foam-templated oleogels? Also, when considering feasibility and industrial scaling up, is it justified to include a process in which one must first freeze samples (1 day), lyofilize (2 day), incorporate oil and rest for 1 day? This includes at least 4 days just to produce fat. Please share these thoughts in your conclusion taking into consideration other oil structuring approaches.
Authors’ response: The reviewer raised a very good point. We have been thinking about this as well. Our main goal was to show utilization of pulse proteins in oil structuring. Biopolymer-stabilized oleogels have not shown great success when incorporated into various food applications. Although cakes were successfully baked using these oleogels, quality of the batters and the cakes prepared using shortening was superior to the oleogel-based batters and cakes. So our research was designed to solve this problem. No doubt this value-added novel product would be costlier than the regular fat and other oil structuring approaches. However, there are multiple benefits as well. Inclusion of plant proteins in oil is the most exciting feather of this structured fat. It will also improve nutritional profile of the fat. The protein also helped in creating structure in the cake, provided a modified cake batter preparation is followed, as shown in our work. We did not do any scaling up and economic analysis, so would not discuss that in the manuscript. However, based on the reviewer’s comment, our thoughts are now added in the conclusion. Please see lines 706-8.
Other minor comments:
Line 35: Merge references
Authors’ response: Done. Line 40
Line 39: English check
Authors’ response: fixed. Line 44
Line 98: Simulated? Simulated in terms of what?
Authors’ response: Simulated was not the right word. We have changed it to “matched”. Line 102.
Line 128: Better write oils containing CW and MAG…
Authors’ response: fixed. Line 131
Line 131: remove “was”
Authors’ response: fixed. Line 134
Line 142: Add rpm of the mixer as was done in section 2.2
Authors’ response: added. Line 147
Line 165 and 168: Why after 30 or 5 min? Please clarify
Authors’ response: It should be 5 min after preparation. We have fixed it and removed the repetition. Line 170
Line 181? Why so much sugar?
Authors’ response: It was based on the AACC method.
Line 192: provide rpm
Authors’ response: added. Line 198
Line 211: Please provide reference for this methodology. Why not measure Apparent density instead?
Authors’ response: Specific gravity was more standard for this work. Reference from our work was added. Line 220
Line 288: Why was canola oil added to the shortening?
Authors’ response: We believe it should be line 228. Nile red was soluble in a solid-like shortening. So we had to dissolve it in liquid canola oil and then that minute amount of oil+Nile red into shortening. Now line 234
Line 383: Which peak?
Authors’ response: peak of the oleofoam. A better reference to that is now added. Line 399
Reviewer 2 Report
Mansucrito is well written and may be of interest within the field of food. Numerous techniques are used and experiments are well described for possible replication. The problem with this study is that too many factors are analyzed and sometimes it is difficult to understand the discussion and conclusions well. In addition, indicating that it is an optimization without having carried out, for example, an analysis by the response surfaces or a design of experiments, sounds strange. The introduction makes a good analysis of the state of the art providing numerous current references. In any case, some questions must be answered and some aspects corrected.
- With so little data within the LVR, it is difficult to ensure that frequency sweeps have been carried out in all cases in that range. It is especially clear in the case of 1%MAG. What do the authors think?
- Why is the study time in stability so long but with so few points?
- The scale in microphotographs is impossible to see.
- Flow curves could fit into a better model than power law. I think it is more feasible to remove some point like the first one from the Canola oil sample that is clearly wrong.
Author Response
Mansucrito is well written and may be of interest within the field of food. Numerous techniques are used and experiments are well described for possible replication. The problem with this study is that too many factors are analyzed and sometimes it is difficult to understand the discussion and conclusions well. In addition, indicating that it is an optimization without having carried out, for example, an analysis by the response surfaces or a design of experiments, sounds strange. The introduction makes a good analysis of the state of the art providing numerous current references. In any case, some questions must be answered and some aspects corrected.
Authors’ response: Thanks for your time in reviewing our manuscript. Our main goal was to compare the two pulse proteins-based oleofoam with shortening. As a control we also have oleofoam made without proteins. However, as the original AACC method was a failure, we had to modify the method of cake batter preparation, hence we added that as another control. We have tried again to fix the discussion and conclusion to make it easier to follow.
As per the response surfaces or a design of experiments, all our compositions were based on previously published research, for which we have explained it clearly. Comparisons between the samples were done with proper statistical analysis. So, no complex response surfaces or a design of experiments was needed.
- With so little data within the LVR, it is difficult to ensure that frequency sweeps have been carried out in all cases in that range. It is especially clear in the case of 1%MAG. What do the authors think?
Authors’ response: It is true that it was not possible to do frequency sweep within LVR for all samples in Fig 2. 1% MAG oleofoam did not show any LVR at all. But it was still done for comparison purposes. Interestingly, all the oleofoams displayed higher G' than G" over the entire frequency range studied, even it was not within LVR for 1% MAG. It is now discussed in the manuscript. See line 369-371. Other discussion involving LVR was also improved. See lines 347-8, and 352-3.
- Why is the study time in stability so long but with so few points?
Authors’ response: The most important time for stability was within the first few hours, however we also wanted to know its stability after a long-term storage (30 days) so that it can be used after storage. Hence 30 days data was also shown. However, we did not measure stability in between, hence not many points are shown.
- The scale in microphotographs is impossible to see.
Authors’ response: We have now added better visible scalebars in the figure and the caption. See figure 4.
- Flow curves could fit into a better model than power law. I think it is more feasible to remove some point like the first one from the Canola oil sample that is clearly wrong.
Authors’ response: With such a good fit to Power law, we think it is okay to use Power law (line 566). We have added a new discussion on the low-shear viscosity profile to better explain their behaviour. See line 558-563.
Reviewer 3 Report
It's an Interesting work. The subject is well known, but there are still some problems connected wit the application of oleogels in bakery products. The level of English is high enough. Some papers are missing and should be cited:
Demirkesen I, Mert B. Recent developments of oleogel utilizations in bakery products. Crit Rev Food Sci Nutr. 2020;60(14):2460-2479. doi: 10.1080/10408398.2019.1649243.
The aim of this study was to provide an overview of the functions of shortening in bakery products and of the field of oleogels with special importance on the updates from recent years and their possible applications in bakery products. With the incorporation of oleogels or oleogel/shortening blends, rheological properties of dough/batters as well as physicochemical properties of resulted products may be resembled to those made with shortening. Conversely, the application of this technique had a role on retaining solid-like properties while possesses a healthier fatty acid profile. Very recent study indicated that gradual replacement of shortening with oleogels have potential for partial reduction of saturated fat without chancing physical properties of gluten free aerated products. Thus, the applications of oleogels may also present more alternatives for celiac sufferers' diet.
Mert B, Demirkesen I. Reducing saturated fat with oleogel/shortening blends in a baked product. Food Chem. 2016 May 15;199:809-16. doi: 10.1016/j.foodchem.2015.12.087.
The objective of this study was to investigate the potential application of Candelilla wax (CDW) containing oleogels for partial replacement of the shortening in cookies. Oleogels were prepared with different amounts of CDW and blended with a commercial bakery shortening. After crystallizing the oleogel/shortening blends by using a pilot scale crystallization unit, the blends were evaluated in a cookie formulations. When the shortening was completely replaced with oleogel softer products were obtained compared to liquid oil, but they were harder than the shortening containing products. On the other hand, partial replacement of shortening with oleogels provided much more acceptable dough and cookie characteristics. Results suggest that gradual replacement of shortening with oleogels may be a suitable approach for reduction of saturated fat in short dough products.
PehlivanoÄŸlu H, Demirci M, Toker OS, Konar N, Karasu S, Sagdic O. Oleogels, a promising structured oil for decreasing saturated fatty acid concentrations: Production and food-based applications. Crit Rev Food Sci Nutr. 2018 May 24;58(8):1330-1341. doi: 10.1080/10408398.2016.1256866.
In this review, history, raw materials and production methods of the oleogels and their functions in oleogel quality were mentioned. Moreover, studies related with oleogel usage in different products were summarized and positive and negative aspects of oleogel were also mentioned. Considering the results of the related studies, it can be concluded that oleogels can be used in the formulation of bakery products, breakfast spreads, margarines, chocolates and chocolate-derived products and some of the meat products.
Corrections:
161-162 The microscopy of the oleofoams was performed at least after 24 h of storage in the refrigerator. What was the maximal time before the sample was observed?
257 Cake crumbs were cut into 2 cm cubes...should be 2 cm3.
258 until half height..... should be "at 50% deformation"
340 should be "strain"
Author Response
Interesting work and worth of publishing. The subject is well known, but there are still some problems connected wit the application of oleogels in bakery products. The level of English is high enough.
Some papers are missing and should be cited:
Demirkesen I, Mert B. Recent developments of oleogel utilizations in bakery products. Crit Rev Food Sci Nutr. 2020;60(14):2460-2479. doi: 10.1080/10408398.2019.1649243.
The aim of this study was to provide an overview of the functions of shortening in bakery products and of the field of oleogels with special importance on the updates from recent years and their possible applications in bakery products. With the incorporation of oleogels or oleogel/shortening blends, rheological properties of dough/batters as well as physicochemical properties of resulted products may be resembled to those made with shortening. Conversely, the application of this technique had a role on retaining solid-like properties while possesses a healthier fatty acid profile. Very recent study indicated that gradual replacement of shortening with oleogels have potential for partial reduction of saturated fat without chancing physical properties of gluten free aerated products. Thus, the applications of oleogels may also present more alternatives for celiac sufferers' diet.
Authors’ response: Thanks for suggesting this important review article. It is now added in the paper. Ref 1.
Mert B, Demirkesen I. Reducing saturated fat with oleogel/shortening blends in a baked product. Food Chem. 2016 May 15;199:809-16. doi: 10.1016/j.foodchem.2015.12.087.
The objective of this study was to investigate the potential application of Candelilla wax (CDW) containing oleogels for partial replacement of the shortening in cookies. Oleogels were prepared with different amounts of CDW and blended with a commercial bakery shortening. After crystallizing the oleogel/shortening blends by using a pilot scale crystallization unit, the blends were evaluated in a cookie formulations. When the shortening was completely replaced with oleogel softer products were obtained compared to liquid oil, but they were harder than the shortening containing products. On the other hand, partial replacement of shortening with oleogels provided much more acceptable dough and cookie characteristics. Results suggest that gradual replacement of shortening with oleogels may be a suitable approach for reduction of saturated fat in short dough products.
Authors’ response: As we did not use oleogel for baking (we used oleofoam) and CW-oleogel did not make oleofoam, and we did not bake cookies, this study would not be a suitable reference in our work.
PehlivanoÄŸlu H, Demirci M, Toker OS, Konar N, Karasu S, Sagdic O. Oleogels, a promising structured oil for decreasing saturated fatty acid concentrations: Production and food-based applications. Crit Rev Food Sci Nutr. 2018 May 24;58(8):1330-1341. doi: 10.1080/10408398.2016.1256866.
In this review, history, raw materials and production methods of the oleogels and their functions in oleogel quality were mentioned. Moreover, studies related with oleogel usage in different products were summarized and positive and negative aspects of oleogel were also mentioned. Considering the results of the related studies, it can be concluded that oleogels can be used in the formulation of bakery products, breakfast spreads, margarines, chocolates and chocolate-derived products and some of the meat products.
Authors’ response: Thanks for suggesting this review article. It is now added in the paper. Ref 2.
Corrections:
161-162 The microscopy of the oleofoams was performed at least after 24 h of storage in the refrigerator. What was the maximal time before the sample was observed?
Authors’ response: The microscopy was performed the very next day, 24 hr after preparation. “at least” has been removed. Please see line 168.
257 Cake crumbs were cut into 2 cm cubes...should be 2 cm3.
Authors’ response: It was cube shape crumbs with 2 cm length. If we consider volume, it should be 8 cm3. We kept it as it was.
258 until half height..... should be "at 50% deformation"
Authors’ response: changed. Please see line 265.
340 should be "strain"
Authors’ response: fixed. Please see line 349.
Round 2
Reviewer 2 Report
Taking into account the reviews carried out and the answers given to the questions, I consider that the manuscript has sufficient quality to be published in Foods in its current corrected version.
Author Response
Comments and Suggestions for Authors
Taking into account the reviews carried out and the answers given to the questions, I consider that the manuscript has sufficient quality to be published in Foods in its current corrected version.
Authors' response: We thank you for taking time and care to review our manuscript.